# A *Eucalyptus* Pht1 Family Gene EgPT8 Is Essential for Arbuscule Elongation of *Rhizophagus irregularis*

Xianrong Che,[a] Sijia Wang,[a] Ying Ren,[a] Xianan Xie,[a] Wentao Hu,[a] Hui Chen,[a] Ming Tang[a]

aState Key Laboratory of Conservation and Utilization of Subtropical Agro-Bioresources, Guangdong Laboratory for Lingnan Modern Agriculture, Guangdong Key Laboratory for Innovative Development and Utilization of Forest Plant Germplasm, College of Forestry and Landscape Architecture, South China Agricultural University, Guangzhou, People's Republic of China

**ABSTRACT** The majority of vascular flowering plants can establish arbuscular mycorrhizal (AM) symbiosis with AM fungi. These associations contribute to plant health and plant growth against various environmental stresses. In the mutualistic endosymbiosis, the AM fungi deliver phosphate ($P_i$) to the host root through highly branched hyphae called arbuscules. The molecular mechanisms of $P_i$ transfer from AM fungi to the plant have been determined, which are dominated by AM-specific $P_i$ transporters belonging to the PHOSPHATE TRANSPORTER 1 (Pht1) family within the subfamily I. However, it is unknown whether Pht1 family proteins are involved in other regulations in AM symbiosis. Here, we report that the expression of EgPT8 is specifically activated by AM fungus *Rhizophagus irregularis* and is localized in root cortical cells containing arbuscules. Interestingly, knockdown of EgPT8 function does not affect the *Eucalyptus grandis* growth, total phosphorous (P) concentration, and arbuscule formation; however, the size of mature arbuscules was significantly suppressed in the *RNAi-EgPT8* lines. Heterogeneous expression of *EgPT4*, *EgPT5*, and *EgPT8* in the *Medicago truncatula* mutant *mtpt4-2* indicates that *EgPT4* and *EgPT5* can fully complement the defects of mutant *mtpt4-2* in mycorrhizal $P_i$ uptake and arbuscule formation, while *EgPT8* cannot complement the defective AM phenotype of the *mtpt4-2* mutant. Based on our results, we propose that the AM fungi-specific subfamily I transporter EgPT8 has novel functions and is essential to arbuscule elongation.

**IMPORTANCE** Arbuscular mycorrhizal (AM) formation in host root cortical cells is initiated by exchanges of diffusible molecules, among which $P_i$ uptake is known as the important feature of AM fungi on symbiosis functioning. Over the last two decades, it has been repeatedly proven that most vascular plants harbor two or more AM-specific Pht1 proteins; however, there is no direct evidence regarding the potential link among these $P_i$ transporters at the symbiotic interface. This work revealed a novel function of a structurally conserved protein involved in lateral arbuscule development. In total, we confirmed that three AM-specific Pht1 family proteins are nonredundant in *Eucalyptus grandis* and that EgPT8 is responsible for fungal arbuscule elongation of *Rhizophagus irregularis*.

**KEYWORDS** *Eucalyptus grandis*, arbuscular mycorrhiza fungi, phosphate uptake, arbuscule elongation, EgPT8

Address correspondence to Ming Tang, tangming@scau.edu.cn, or Hui Chen, chenhui@scau.edu.cn.

The authors declare no conflict of interest.

The arbuscular mycorrhizal (AM) symbioses formed by vascular plants and AM fungi have existed for more than 400 million years, and >70% of vascular plants live in close association with the ubiquitous mutualistic symbioses in terrestrial ecosystems (1–3). In the process of establishing a symbiotic relationship, extraradical hyphae, which originate from soil-germinating spores, penetrate the epidermal cells of the host root, cross the outer root cell layers, elongate within the intercellular spaces, and eventually invade cortical cells, where they form highly branched tree-like arbuscules (4, 5). Arbuscules are enveloped by the periarbuscular membrane (PAM), a specialized host plasma membrane, within the inner cortical cells to separate the fungus from the plant

cell cytoplasm (6–8). AM fungi are considered among the dominant components of the plant microbiota, the biodiverse ecosystem of microbial communities that are closely associated with multicellular individuals and impact plant health (9–11).

In the symbiotic association, organic carbon fixed by photosynthetic hosts is transferred to the AM fungi in exchange for inorganic mineral nutrients, in particular inorganic orthophosphate ($P_i$), an essential macronutrient for plants (12–15). In AM, plants have two pathways for $P_i$ absorption from soil: either directly via root epidermal cells or indirectly via mycorrhizal cortical cells at the symbiotic interface (16, 17). During the AM $P_i$ uptake pathway, $P_i$ is absorbed by extraradical hyphal networks and transferred through the AM fungal hyphae as polyphosphate (polyP), subsequently, polyP is hydrolyzed via arbuscule, and free $P_i$ is exported from the fungus to the periarbuscular space (PAS) (18–20). The import of $P_i$ across the PAS into the host cortical cells is mediated by AM-inducible phosphate transporters of the Pht1 family, which are located on the PAM (21–23).

Since the first AM-associated PHT1 family transporter StPT3 (hereafter, the short-form PT is used to refer to all Pht1 genes and proteins) is reported to function in mycorrhizal potato root, several orthologs have been determined that are related to take up $P_i$ in AM roots (12, 18, 22–24). Now it is clear that in all AM plants, there is at least one AM-specific Pht1 family member acting exclusively, or predominantly, during AM symbiosis and is grouped in subfamily I, such as SbPT1 (*Sorghum bicolor*), SiPT9 (*Setaria italica*), BdPT11 (*Brachypodium distachyon*), HvPT11 (*Hordeum vulgare*), OsPT11 (*Oryza sativa*), and ZmPT6 (*Zea mays*) (25–29).

Further studies showed that these genes belonging to subfamily I can be divided into two subgroups: PHT1;4 and PHT1;8 (23). In subgroup PHT1;4, the functions of most of the proteins have been revealed. In tomato, two AM-specifically induced members of subfamily I, SlPT4 and SlPT5, are functionally redundant (30). *Medicago truncatula* harbors two proteins belonging to subfamily I, while only MtPT4 was functionally characterized in arbuscular mycorrhizal roots (21). In *Astragalus sinicus*, although both AsPT1 and AsPT4 are responsible for arbuscular formation, mycorrhizal $P_i$ uptake is not mediated by AsPT1 (23). However, little is known about the function of PHT1;8 subgroup proteins.

*Eucalyptus* species, belonging to the *Myrtaceae* family, are the main hardwood and fiber sources worldwide, due to their ability to grow quickly (10- to 15-year rotation periods) (31). Moreover, eucalypts have AM fungi and ectomycorrhizal (ECM) fungi associated with their roots; these fungi can improve tree growth in nutrient-deficient soils, particularly by increasing the absorption of $P_i$ (32–35). *Eucalyptus grandis* harbors three AM-specific subfamily I transporters, but their roles in AM symbiosis are unknown.

In this study, we employed a dual strategy: (i) to characterize the AM formation of *E. grandis* transformed plants for which *EgPT8* is knocked down and (ii) to carry out cross-species complementation tests. Surprisingly, we did not find redundant regulation of *E. grandis* AM symbiosis by three members of the Pht1 gene family. Instead, we confirmed that EgPT8 is not responsible for the transport and uptake of $P_i$ and also is not involved in AM formation, while EgPT8 plays a key role in arbuscule elongation of *Rhizophagus irregularis*.

## RESULTS

**Phylogenetic tree and conservativeness analysis of PHT1 family proteins.** In our previous research, we reported that multiple $P_i$ transporters of the Pht1 family were recruited for AM symbiosis in *E. grandis* (36). To predict the functional differences of these genes, a neighbor-joining tree was constructed using a multiple sequence alignment of *E. grandis* Pht1 proteins and sequences from other plant $P_i$ transporters. As shown in Fig. S1, except for rice OsPT13 as the outgroups, Pht1 proteins clustered into six subfamilies. Interestingly, *E. grandis* harbors three nonorthologous $P_i$ transporters, EgPT4, EgPT5, and EgPT8, which belong to the AM-specific subfamily I. EgPT4 and EgPT5 appeared to be closely grouped with several AM-specific Pht1 homologs that

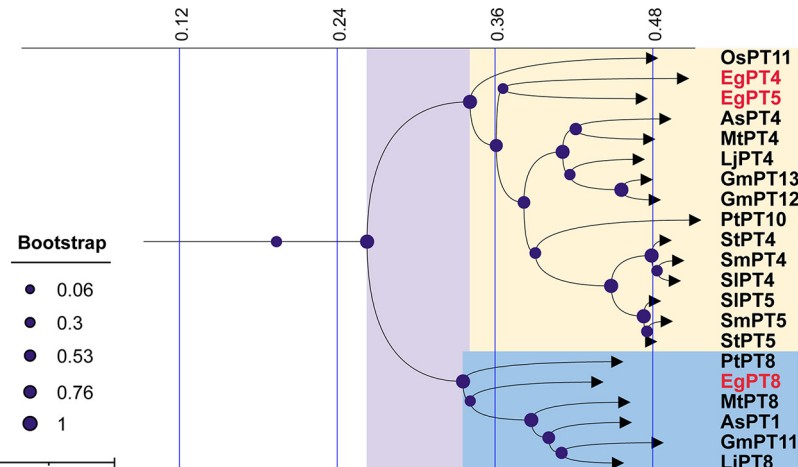

**FIG 1** Phylogenetic analysis of PHT1 subfamily I genes. An unrooted phylogenetic tree was divided from Fig. S1. Subfamily I is composed of two subgroups PHT1;4 (yellow shading) and PHT1;8 (blue shading). The numbers above the diagram indicate tree scale. The transporters and corresponding plant species are *Eucalyptus grandis*, EgPT4, EgPT5, and EgPT8; tomato (*Solanaceae lycopersicum*), SlPT4 and SlPT4; potato (*Solanaceae tuberosum*), StPT4 and StPT5; eggplant (*Solanaceae melongena*), SmPT4 and SmPT5; *Medicago truncatula*, MtPT4 and MtPT8; *Lotus japonicus*, LjPT4 and LjPT8; soybean (*Glycine max*), GmPT11, GmPT12, and GmPT13; rice, OsPT11; *Astragalus sinicus*, AsPT1 and AsPT4; and *Populus trichocarpa*, PtPT8 and PtPT10. Accession numbers are given in Table S1 in the supplemental materials.

have been reported to function as $P_i$ uptake transporters, such as the MtPT4, SlPT4, OsPT11, and AsPT4 (Fig. 1) (22, 23, 30, 37). However, EgPT8 belonged to subgroup Pht1;8 and is related to MtPT8, GmPT11, LjPT8, PtPT8, and AsPT1. Among them, the expression of AsPT1 is highly induced in AM roots, while the expression of PtPT8, a $P_i$ transporter from tree species, is not induced in any of the poplar mycorrhizal tissues (23, 38). These findings prompted us to determine whether EgPT8 is required for the AM symbiosis and is involved in the AM-mediated uptake of $P_i$.

By comparing the genomic DNA and cDNA sequences, *EgPT8* was found to contain a 1,656-bp-long open reading frame (ORF) without introns (Fig. S2). The predicted crystal structure suggests that EgPT4, EgPT5, and EgPT8 are highly conserved compared with the $P_i$ transporter 4J05.1 (Fig. S3A to D) (39). Overall comparison of the 3D structure of the well known $P_i$ transporter MtPT4 and the model structure of EgPT4, EgPT5, and EgPT8 revealed a high level of superposition (21). Furthermore, the model structure of EgPT4, EgPT5, and EgPT8 suggest the presence of 12 transmembrane helices disposed in a similar orientation as MtPT4, indicating that their protein structure is conserved. To sum up, these results suggest that despite the conservation of the structure of EgPT4, EgPT5, and EgPT8, they may have different functions due to their different phylogenetic subgroups.

**Relative expression and subcellular localization analysis of EgPT8 in AM fungi-colonized tissues.** To gain an overview of *EgPT8* responses to AM fungal colonization, different experimental groups were set up to detect the expression pattern of the Pht1;8 subgroup $P_i$ transporter. First, real-time PCR was performed to investigate the expression of *EgPT8* in *E. grandis* tissues colonized by *R. irregularis* under low $P_i$ conditions. *E. grandis* tissues of root and shoot inoculated or mock-inoculated with AM fungi were harvested at 6 weeks after inoculation. As expected, the expression levels showed that *EgPT8* is specifically induced in the AM root tissue (Fig. 2A). Next, a time course treatment from 14 to 49 days postinoculation (dpi) was carried out to determine the expression level of *EgPT8* in *E. grandis* roots at different time points of AM symbiosis. The induction rate continuously increased with the duration of symbiotic treatment and peaked at 35 dpi, while the transcription level of *EgPT8* was low at 42 dpi (Fig. 2B). Subsequently, the expression of *EgPT8* was also examined in AM *E. grandis* roots in response to different $P_i$ stress. As shown in Fig. 2C, *EgPT8* was strongly induced at low $P_i$ conditions. However, the induction of *EgTP8* was not observed in the high $P_i$ supplement (3,000 $\mu$M) condition. Furthermore,

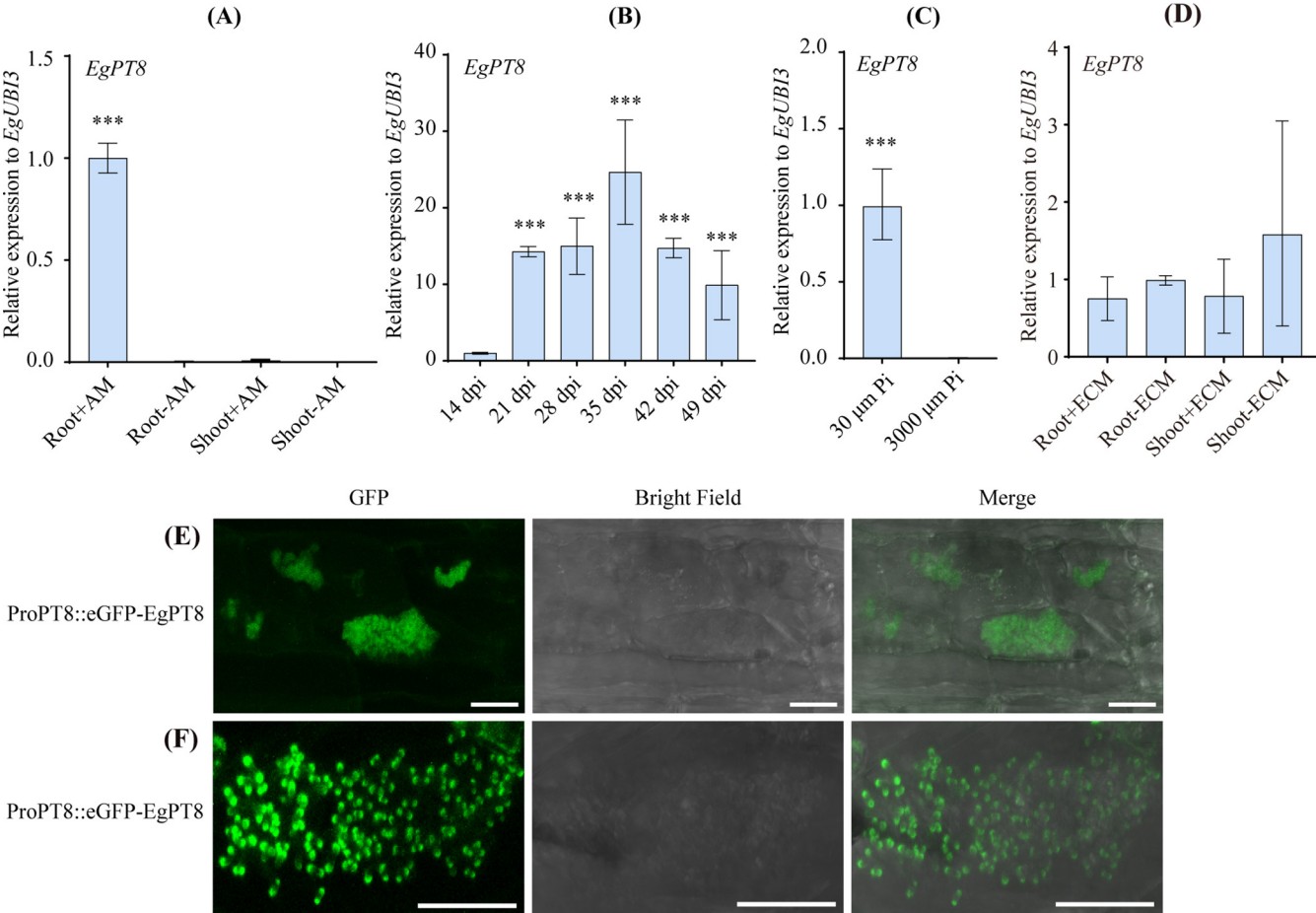

**FIG 2** Spatiotemporal expression patterns and subcellular localization of *EgPT8*. (A) Quantification of transcriptional levels of *EgPT8* in the different mycorrhizal (+AM) and nonmycorrhizal (−AM) tissues of *E. grandis* seedlings grown for 49 days under low $P_i$ (30 $\mu$M) conditions. The Shoot-AM was set as the control. (B) Real-time reverse transcription (RT)-PCR analysis of *EgPT8* expression in mycorrhizal *E. grandis* roots at 14, 21, 28, 35, 42, and 49 days postinoculation (dpi) with *Rhizophagus irregularis*; 14 dpi was set as the reference group. (C) Expression *EgPT8* in response to $P_i$ availability. Plants materials were cultured under low (30 $\mu$M) or high (3,000 $\mu$M) $P_i$ conditions and harvested 49 days after inoculation. (D) Quantitative reverse transcription (qRT-PCR) of *EgPT8* was detected in tissues inoculated (+ECM) or noninoculated (−ECM) with ECM fungus *Scleroderma bovista*. *S. bovista* was inoculated into 6-month-old *E. grandis* seedlings and cultured for 3 months. Root+ECM was set as the reference groups. (E, F) Subcellular localization analysis of EgPT8 in AM *M. truncatula* roots. Laser-scanning confocal microscope images of the mycorrhizal *M. truncatula* hairy root expressing ProEgPT8::eGFP::EgPT8. Bars, 20 $\mu$m. The data are shown as the means ± standard deviation of three biological replicates (*n* = 3). ***, *P* < 0.001 (Student's *t* test). AM, arbuscular mycorrhizal; ECM, ectomycorrhizal; eGFP, enhanced green fluorescent protein.

after 6 months of symbiotic growth with ECM fungus *Scleroderma bovista*, qualitative RT-PCR was conducted in different *E. grandis* tissues inoculated or noninoculated with *S. bovista*. The result showed that the expression of *EgPT8* was not upregulated (Fig. 2D).

It has been repeatedly proven that symbiotic $P_i$ transport from AM fungi into plant cells closely relies on the transporters located on the PAM (12, 22). Therefore, to determine the subcellular localization of EgPT8, its CDs (coding sequence) was fused in-frame with the 3′ end of the green fluorescent protein (GFP) reporter gene driven by the 35S cauliflower mosaic virus promoter (35::GFP-EgPT8). The construct was then transformed into the *Nicotiana benthamiana* leaf epidermal cells. As shown in Fig. S4, eGFP-EgPT8 fusion was colocalized with the endoplasmic reticulum (ER) marker HDEL-mCherry. This indicates that EgPT8 was resident in the ER of *N. benthamiana* leaf epidermal cells in nonmycorrhizal tissue.

To further explore the precise subcellular localization of EgPT8, the full-length CDs of *EgPT8* with the GFP reporter gene was amplified from the fusion of 35::GFP-EgPT8 and reconstructed into the modified binary vectors pCAMBIA1305S under the control of its native promoter to generate the resulting construct of ProPT8::GFP-EgPT8. This construct was transformed into *M. truncatula*. As expected, in *M. truncatula* hairy roots

colonized by *R. irregularis*, the ProPT8::GFP-EgPT8 fusion protein showed specific localization in cells containing arbuscules (Fig. 2E and F). This evidence indicates that EgPT8 is localized in AM fungi-colonized cortical cells, which is similar to the earlier reporters related to the PAM-localized nature of AM-specifically induced $P_i$ and $NO_{3-}$ transporters, MtPT4, AsPT4, and OsNPF4.5 in *M. truncatula*, *A. sinicus*, and rice (23, 40, 41). These findings also lend evidence to support that the localization of subfamily I proteins is conserved in tree species. To sum up, our findings confirm that the expression *EgPT8* is specific in cells containing arbuscules.

**Function of EgPT8 in yeast cells.** The phosphate transport capacity of EgPT8 was evaluated by heterologous expression in the five inorganic phosphate transporter-defective *S. cerevisiae* mutant strain EY917 (42). The wild-type yeast strain EY57 was used as a positive control. Unexpectedly, EgPT8 cannot restore the growth defect of EY917 when the organic phosphate (glycerol-3-phosphate) was not provided in the medium. In contrast, EgPT4 can complement the growth of EY917 (Fig. 3A). To ensure the growth of mutant yeast EY917, 1 mM glycerol-3-phosphate was applied. Surprisingly, EgPT8 restored the $P_i$ uptake capability in mutant EY917, while EgPT4 and EgPT5 complemented the EY917 more efficiently than EgPT8 (Fig. 3B). The results demonstrate that EgPT8 was able to take up $P_i$ and transport it into yeast cells.

To determine the subcellular localization of EgPT8 in yeast cells, the full-length coding sequence of *EgPT8* was fused to the N-terminal GFP fusion vector pUG36 and transformed the resulting plasmid pUG36-GFP-EgPT8 into the wild-type yeast strain EY57. As expected, the observed GFP signal showed that EgPT8 was located in the plasma membrane (PM) of yeasts (Fig. 3C), similar to the localization pattern of EgPT4, EgPT5, and ScPho84.

Next, we introduced pFL61-EgPT8 into EY917 to test the growth tendency. In agreement with the complementation of EgPT8 in EY917, the growth rate of mutant EY917 carrying pFL61-EgPT8 is higher than that of EY917 cells with pFL61 or pFL61-EgPT5 at the logarithmic phase (from 8 to 20 h; Fig. 3D) but is lower than EgPT4 expressing in EY917.

Moreover, the total P concentration in the EY57 and EY917 was confirmed. As expected, the EY917 yeast cells harboring pFL61-EgPT8 showed a significant increase in total P uptake over the empty vector expressed in the EY917 strain (Fig. 3E). In agreement with the complementation and growth curve of EgPT8 observed in mutant strain EY917 (Fig. 3B and D), the data from $P_i$ concentrations indicated that EgPT8 can absorb $P_i$ in the medium to facilitate the yeast growth. These results demonstrate that EgPT8 functions as a $P_i$ transporter and is localized in the plasma membrane when expressed in yeast cells.

**Growth phenotype of *EgPT8-RNAi* lines inoculated with AM fungi.** To evaluate the function of EgPT8 in AM symbiosis, we knocked down EgPT8 using an RNA interference (RNAi) strategy (43). An expression cassette, containing fragments of the *EgPT8* specific interference region, was cloned in the pK7GWIWG2D(II)-RootRed vector. The BLASTn results also did not reveal any off-target base-pairing of *EgPT8* with other parts of the eight *E. grandis* PHT1 genomes and thus reduced the chances of off-site silencing (Fig. S5). Following the *Agrobacterium rhizogenes*-mediated hairy root transformation method of *E. grandis* (42), we obtained several transgenic lines that were screened by the DsRed reporter gene using the fluorescence microscope (Fig. S6). When the *EgPT8-RNAi* lines were harvested at 49 dpi under low $P_i$ conditions, we identified three lines by qRT-PCR determination, *EgPT8-RNAi-1*, *EgPT8-RNAi-2*, and *EgPT8-RNAi-3*, showing significantly decreased expression levels of *EgPT8* (Fig. 4B).

Interestingly, the growth of the control seedlings and *EgPT8-RNAi* lines showed a similar phenotype (Fig. 4A). In agreement with the growth presentation, there was also no significant difference in the biomass of root and shoot tissue between the *EgPT8-RNAi* and control lines under 30 $\mu$M $P_i$ conditions (Fig. 4C and D). Since we have confirmed that EgPT8 functions as a $P_i$ transporter in yeast cells (Fig. 3E), we determined the total P concentration in symbiotic roots and shoots of *EgPT8-RNAi* and control lines. Unexpectedly, the total P concentrations in AM roots and shoots of the *EgPT8-RNAi*

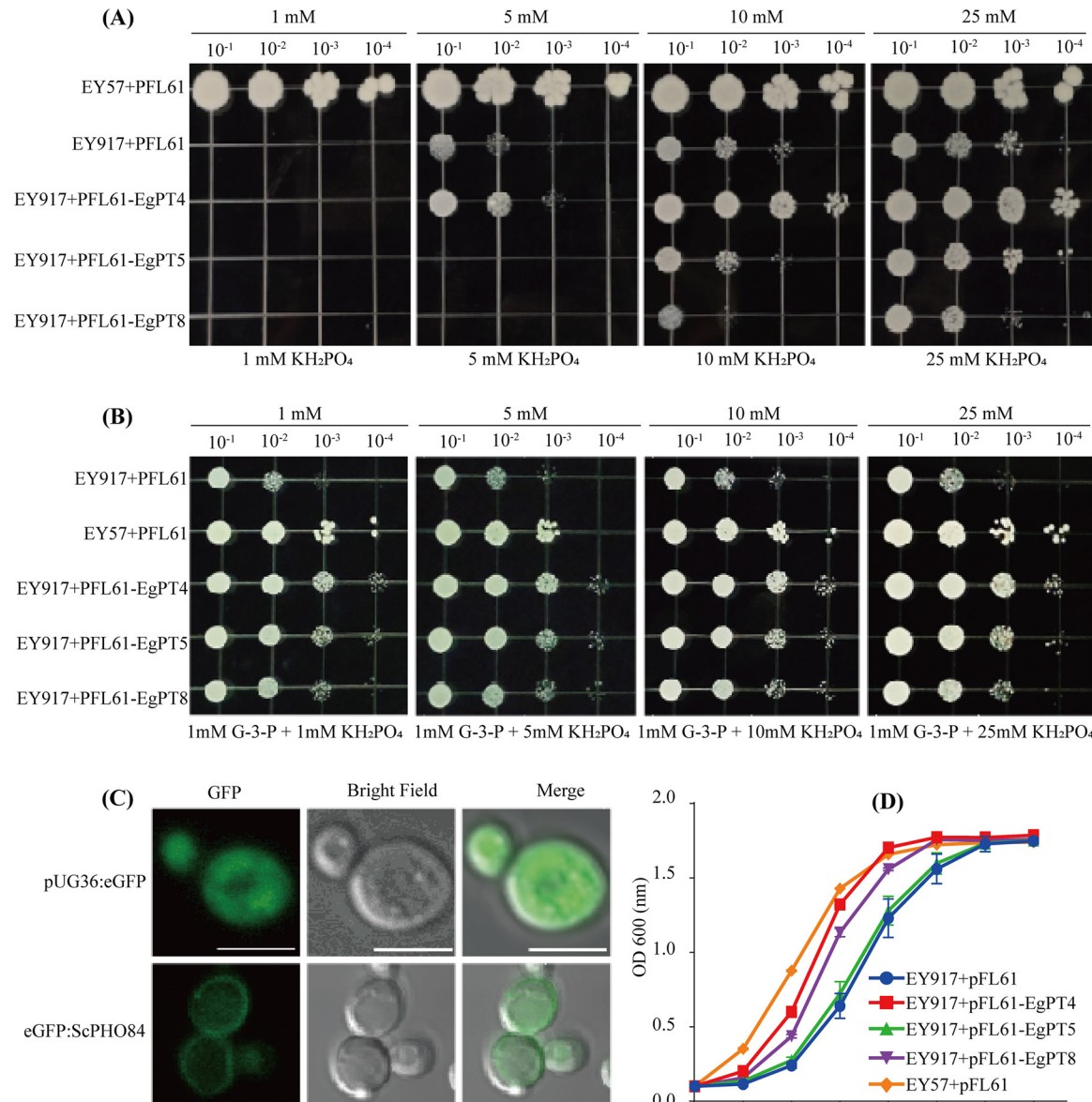

**FIG 3** Functional characterization of EgPT4, EgPT5, and EgPT8 in the yeast mutant strain EY917. (A, B) Functional complementary analysis of the yeast mutant EY917 (Δ*null*) defective in five inorganic phosphate transport genes expressing *EgPT4*, *EgPT5*, and *EgPT8*.

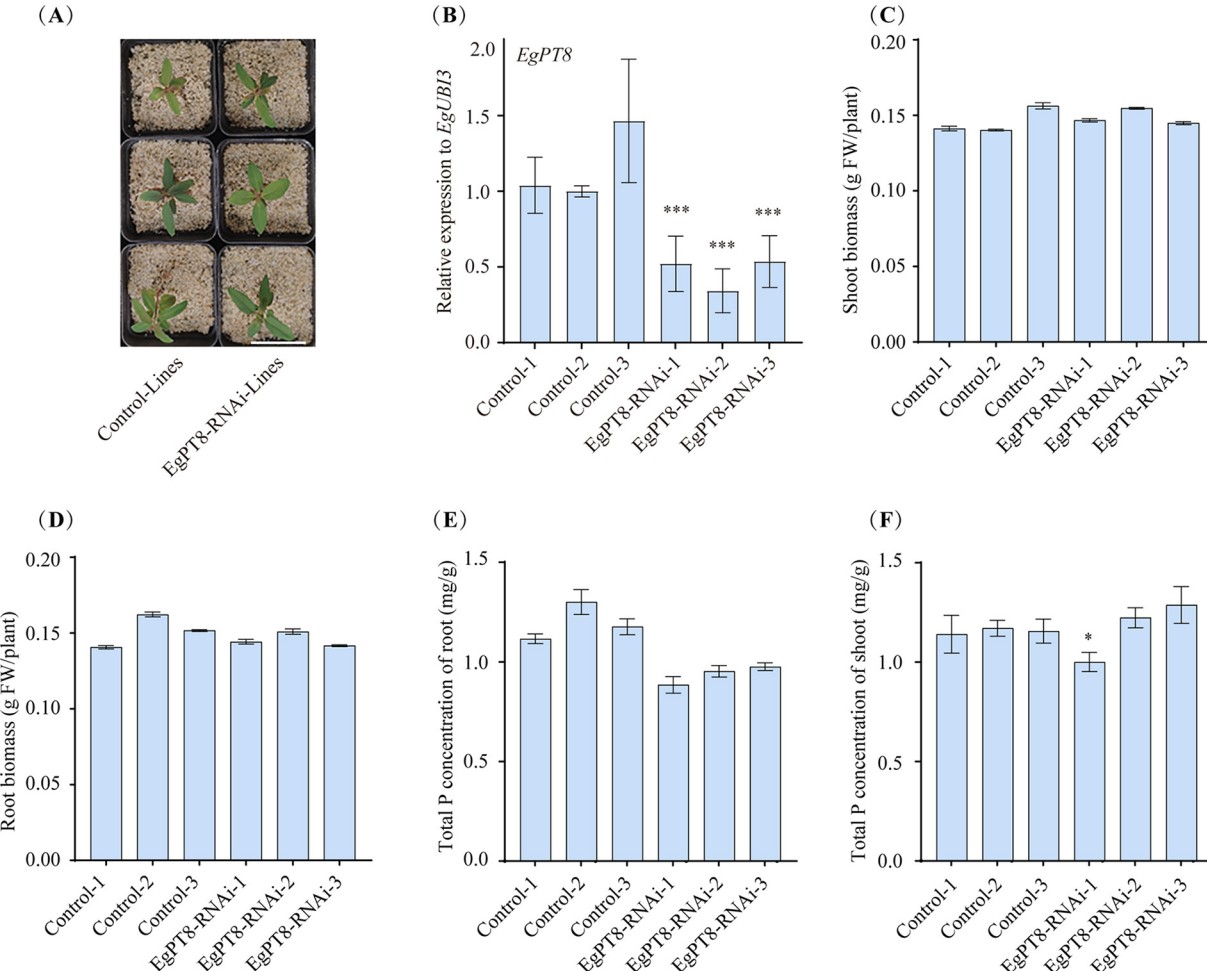

**FIG 4** Effects of interference of *EgPT8* function on plant growth and P$_i$ uptake in the presence of AM fungal colonization. The control and *EgPT8*-RNAi plants were inoculated with *R. irregularis* and grown under 30 $\mu$M P$_i$ supply conditions for 7 weeks. (A) The shoot growth performances of wild-type and *EgPT8*-RNAi seedlings. Bar, 5 cm. (B) The relative expression of EgPT8 in control and *EgPT8*-RNAi lines. (C, D) Shoot and root biomass (fresh weight [FW]). (E, F) Total P concentration of shoot and root. The error bars indicate standard deviation ($n = 3$). Controls 1, 2, and 3 were set as the reference groups. *, $P < 0.05$, ***, $P < 0.001$ (Student's *t* test).

lines showed no significant difference (Fig. 4E and F) compared with the controls. These findings indicate that EgPT8 is dispensable for absorbing P$_i$ from *R. irregularis*.

**Arbuscule formation of *R. irregularis* in symbiotic *RNAi-EgPT8* lines.** An earlier study has determined that *AsPT1* is required for AM symbiosis (23); as *EgPT8* and *AsPT1* were grouped in the same subgroup (Fig. 1), we qualified the mycorrhization in the RNAi and control lines. When the function of *EgPT8* was knocked down, the total fre-

**FIG 3** Legend (Continued)
The transformed yeast strains were incubated in YNB (−Ura) medium with different P$_i$ concentrations at pH 5.8 to an optical density at 600 nm (OD$_{600}$) of 0.5. An equal volume of 10-fold serial gradient dilutions was added to YNB (−Ura) agar plates containing different combinations of P as described above. The spotted plates were incubated at 28°C for 6 days (A) or 3 days (B). (C) Subcellular localization of EgPT4, EgPT5, and EgPT8 in wild-type yeast strain EY57. Confocal laser-scanning microscope images of wild-type or mutant yeast cells carrying either pFL61 or pFL61-*EgPT4*, pFL61-*EgPT5*, pFL61-*EgPT8*. The left panels, middle panels, and right panels indicate GFP fusion, bright-field, and merged images, respectively. The pUG36 expressing GFP reporter gene is cytoplasmic control, while the ScPHO84 phosphate transporter expressed in pUG36 presents as a plasma membrane marker. Bars, 10 $\mu$m. (D) Growth curves of transformed yeast cells overexpressing *EgPT4*, *EgPT5*, or *EgPT8*. The positive strains were cultured in YNB (-Ura) medium provided with 1 mM glycerol-3-phosphate and 1 mM inorganic phosphate at 28°C for 28 h. The data are shown as the means ± SD of three biological replicates ($n = 3$). (E) Phosphate uptake into yeast cells with empty vector and expressing *ScPHO84*, *EgPT4*, *EgPT5*, or *EgPT8*. The yeasts EY917 and EY57 were cultured in YNB (−Ura) medium containing 1 mM P$_i$ (K$_2$HPO$_4$) and 1 mM glycerol-3-phosphate. Yeast cells were collected when the OD$_{600}$ was 1.0. All data shown are averages. The error bars represent standard deviation values ($n = 3$). The EY917+PFL61 group was set as the reference group. *, $P < 0.05$; ***, $P < 0.001$ (Student's *t* test).

quency (F%), mycorrhizal intensity (M%), and arbuscular abundance (A%) in control roots showed no significant differences compared to the EgPT8-RANi line (Fig. 5A). Furthermore, the expression of five AM-associated genes in *E. grandis* was performed in the roots of control and RNAi lines to evaluate whether the knockdown of EgPT8 affects their expression. We did not gain any differences between RNAi and control lines according to the expression patterns in Fig. 5B to E.

AM-inducible plant PHT1 family $P_i$ transporters localized in the PAM are responsible for the $P_i$ uptake from the PAS; once the formation of arbuscules in symbiotic roots were blocked, the free $P_i$ within the intraradical mycelium (IRM) could not be transported to plant cells through the symbiotic interface, resulting in the accumulation of polyphosphate (polyP) in AM fungal vesicles (44–46). Therefore, to further investigate whether the knockdown of *EgPT8* by RNAi affects $P_i$ transport in intraradical hyphae, we performed the expression analysis of the genes involved in the PHO pathway of *R. irregularis* in control and RNAi lines through qRT-PCR (Fig. S7). The results showed that the expression patterns of the polyP synthesis-related genes (*VTC1* and *VTC2*) and polyP metabolisms-related genes (*RiPPN1* and RiPPX1) have no difference in transformed AM roots of *EgPT8-RNAi* lines and control seedlings (Fig. S7A to D). These findings are in concordance with the expression patterns of AM-induced genes in *E. grandis* (Fig. 5B to E) and suggest that a loss of *EgPT8* function does not affect the $P_i$ transport in symbiotic roots.

**Arbuscule development in control and RNAi lines.** The results of mycorrhization and the expression of AM-induced genes in RNAi lines demonstrated that *EgTP8* is not responsible for the $P_i$ uptake and AM symbiosis. Thus, to further support this evidence, we assess the arbuscule development in control and RNAi lines. As expected, the phenotype of intraradical hyphae and arbuscule in normal *EgPT8-RNAi* lines was similar to control seedlings (Fig. 6A). To further evaluate the arbuscule development, 100 mature arbuscules were randomly selected, and the length and area of individual arbuscules were measured in *EgPT8-RNAi* and control roots (Fig. 6B and C). Interestingly, the length and area of arbuscules in *EgPT8*-RNAi lines were significantly depressed compared to the controls, indicating that the silence of EgPT8 hampered the growth of arbuscules inside the roots.

**Complement the defective AM phenotype of *mtpt4-2* mutant.** To demonstrate whether the AM-specific genes of *E. grandis* were able to perform similar functions to its orthologs in AM symbiosis, the CDS of *EgPT4*, *EgPT5*, and *EgPT8* under the control of their native promoters were introduced into the *mtpt4-2* mutant plants. The transformed lines, *ProEgPT4::eGFP::EgPT4*, *ProEgPT5::eGFP::EgPT5*, and *ProEgPT8::eGFP::EgPT8*, as well as the corresponding WT and *mtpt4-2* mutant plants, were inoculated with *R. irregularis*, and cultured under low $P_i$ conditions for 6 weeks. Under these conditions, the symbiotic shoot and root biomass showed no significant difference among the three transgenic lines, as well as the WT and *mtpt4-2* mutant plants (Fig. S8A and B). Further observation of the total P concentration in AM roots revealed no significant differences in WT, transgenic lines, and mutant lines (Fig. S8C). In contrast, in WT, *ProEgPT4::eGFP::EgPT4*, and *ProEgPT5::eGFP::EgPT5* lines, the detected total P concentration in AM shoots is significantly higher than those in the *ProEgPT8::eGFP::EgPT8* and mutant *MtPT4-2* plants (Fig. 7A), indicating that EgPT4 and EgPT5 have the function of symbiotic $P_i$ uptake in *E. grandis*, while EgPT8 is not responsible for $P_i$ uptake from AM fungi.

To further confirm the colonization levels, total frequency (F%) of mycorrhization, mycorrhizal intensity (M%), and arbuscule abundance (A%) in transgenic roots were determined after staining with WGA488 (Fig. 7B). As expected, *ProEgPT4::eGFP::EgPT4* and *ProEgPT5::eGFP::EgPT5* lines showed high levels of F%, M%, and A% compared to the mutant *mtpt4-2* and *ProEgPT8::eGFP::EgPT8* lines, which were similar to the WT. However, the mycorrhizal levels in *ProEgPT8::eGFP::EgPT8* transgenic roots did not differ significantly from those in the mutant *mtpt4-2*. In addition, the phenotype in transgenic mycorrhizal roots was observed (Fig. 7C to G). Abundant well developed arbuscules could be observed in the mycorrhizal roots of WT and *mtpt4-2* plants expressing the eucalypts *EgPT4* and *EgPT5* genes (Fig. 7C, E, and F). However, the arbuscule

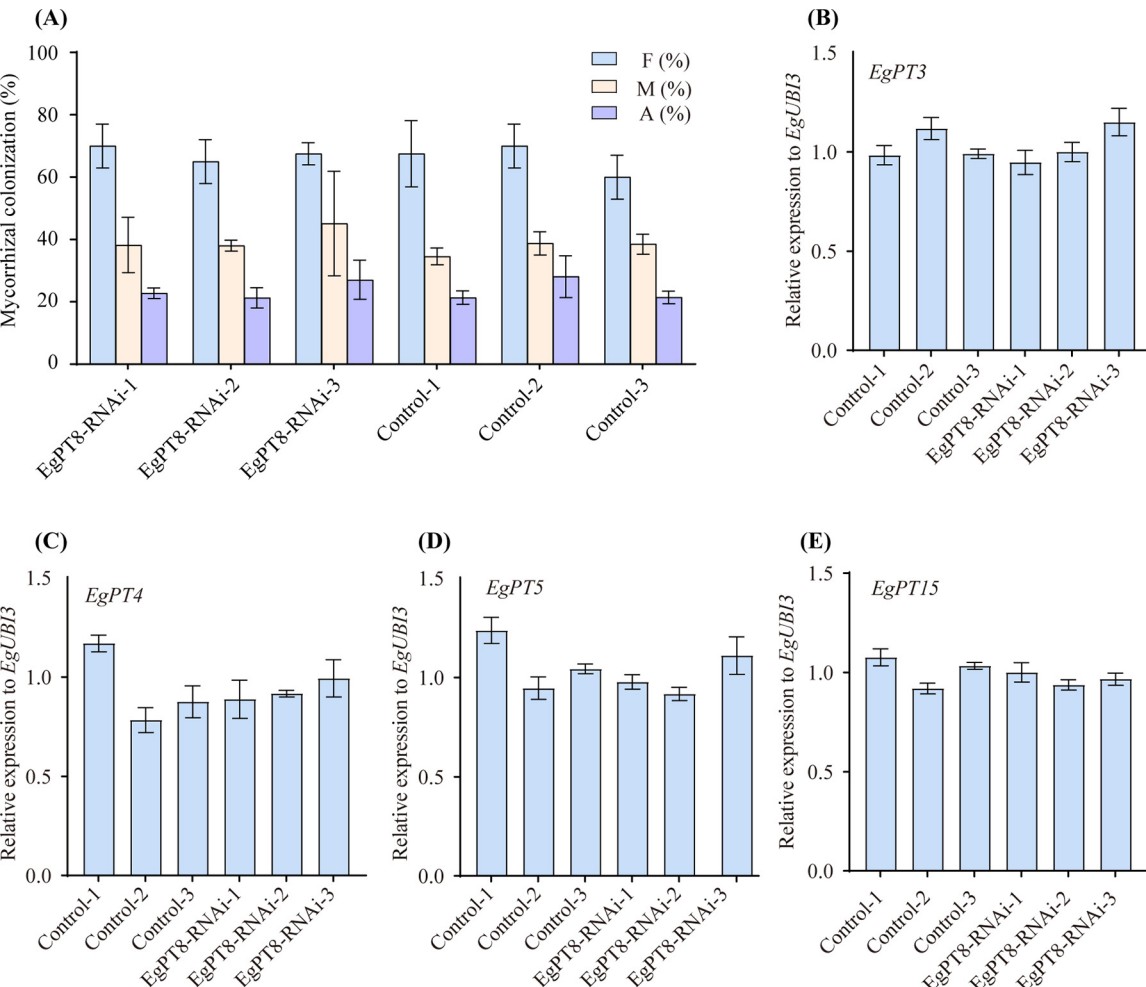

**FIG 5** Mycorrhizal colonization and expression analyses of control and *EgPT8*-RNAi lines. (A) Total mycorrhizal frequency (F%), mycorrhizal intensity (M%), and arbuscule abundance (A%) in mycorrhizal roots of both the WT and *EgPT8*-RNAi mutant plants were determined. (B to E) Relative expression levels of four mycorrhizal-inducible PHT1 family genes in the root of control and *EgPT8*-RNAi lines during AM symbiosis. Error bars indicate SD ($n = 3$; Student's *t* test). Controls 1, 2, and 3 were set as the reference groups.

formation in *ProEgPT8::eGFP::EgPT8* lines was restricted, and morphological characteristics were identical to the *mtpt4-2* mutant. These findings highlight that *EgPT4* and *EgPT5* can fully complement the defects of the *mtpt4-2* in AM colonization, arbuscule formation, and mycorrhizal P uptake, whereas *EgPT8* cannot restore the AM phenotype of the mutant *mtpt4-2*, nor can it assist AM symbiosis to absorb $P_i$ from AM fungi.

## DISCUSSION

Our findings provide proof that EgPT8, an AM-specific subfamily I protein, is mainly expressed in AM roots and can restore the growth defects of mutant yeast EY917. In addition, we confirm that the loss function of *EgPT8* does not affect the formation of arbuscule nor the $P_i$ uptake in symbiotic AM *E. grandis*; however, the length and area of arbuscules are significantly depressed in *RNAi-EgPT8* lines, suggesting that *EgPT8* is essential for arbuscule elongation. Moreover, *EgPT8* cannot complete the defective phenotype of the mutant *mtpt4-2*, indicating that EgPT8 is not a key factor during the formation of arbuscules.

**Conserved subfamily I PHT1 proteins differ in their mechanism of response to the AM fungi.** The $P_i$ transporters of PHT1 family, first isolated and characterized in *Arabidopsis*, perform key functions in the uptake and transport of $P_i$ in plants (47). Although the first reported AM-associated PHT1 member is StPT3, phylogenetic tree analysis demonstrates that it was not clustered with $P_i$ transporters from subfamily I, which harbored the AM-specific $P_i$ transporters from both monocot and dicot species (30). The

**(A)**

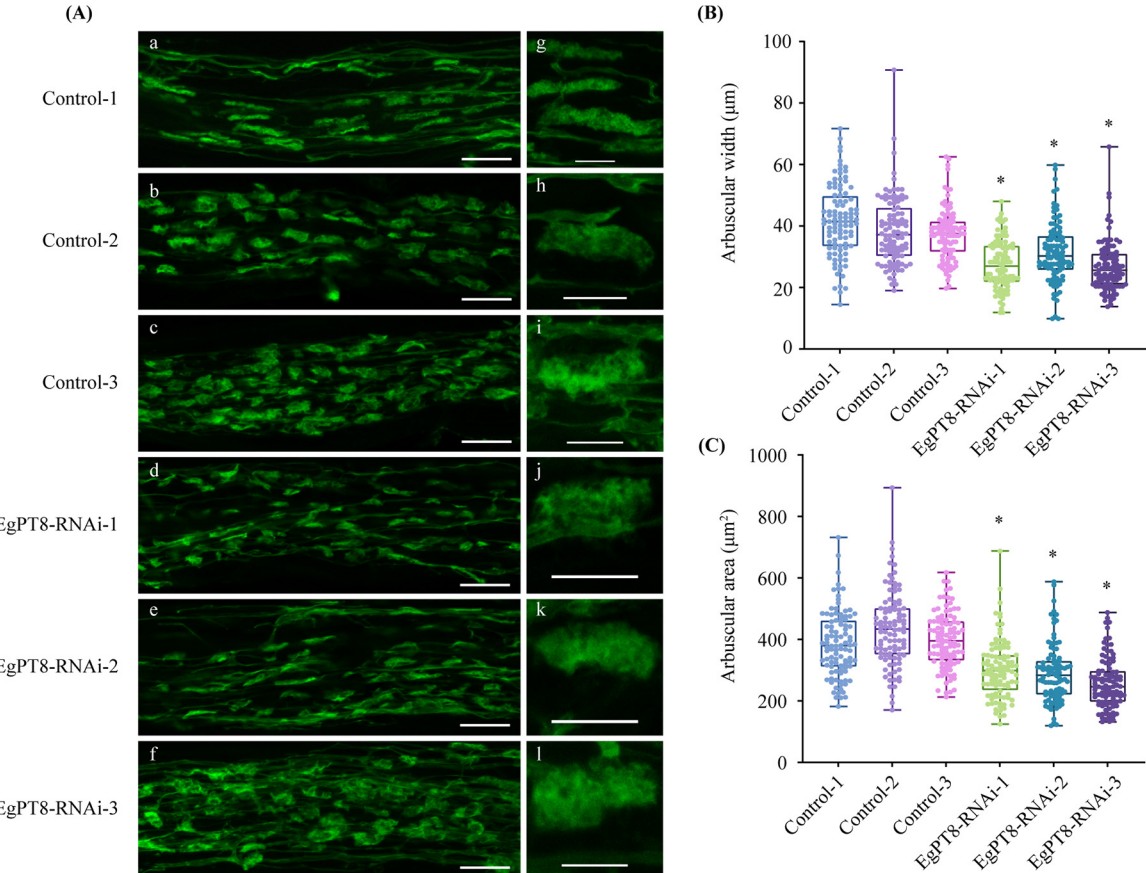

**FIG 6** Mycorrhizal phenotypes and physical characteristics of control and *EgPT8*-RNAi lines. (A) Arbuscular phenotype in control and RNAi lines. *E. grandis* mycorrhizal roots were labeled with WGA 488 dye. Confocal images of *R. irregularis* within the control (a to c) and *EgPT8*-RNAi (d to f) roots of *E. grandis*, and the magnified images (×63 water objective) show the arbuscule morphology in control (a to c and g to i) and *EgPT8*-RNAi (d to f and j to l) roots. Bars, 20 μm (a to f) and 50 μm (g to l). (B, C) Analysis of arbuscule width and area. A total of 100 arbuscules were randomly selected for each sample. Data measurement is based on the software ImageJ (https://imagej.nih.gov/ij/). Controls 1, 2, and 3 were set as the reference groups. Asterisks indicate a statistically significant difference: *, $P < 0.05$ (Student's *t* test).

*E. grandis* harbors three AM-activated $P_i$ transporters belonging to subfamily I (Fig. S1); their 3D structures showed conserved features with the MtPT4 (Fig. S3), an *M. truncatula* member from subfamily I (18). However, as shown in Fig. 1, EgPT4 and EgPT5 closely grouped with the MtPT4, AsPT4 in the PHT1;4 subgroup, while EgPT8 closely clustered together with the legume proteins AsPT1and MtPT8 within the PHT1;8 subgroup (20, 48), suggesting that subfamily I transporters may have different functions.

Members of subgroup PHT1;4, such as MtPT4 and AsPT4, have been characterized as $P_i$ uptake proteins in symbiotic roots inoculated with AM fungi and are essential for AM symbiosis (18, 20). In tomato, two subgroup PHT1;4 proteins, SlPT4 and SlPT5, demonstrated considerable redundancy upon knockout of SlPT4 (27). Although most monocot plants contain only one AM-specific $P_i$ transporter belonging to subgroup PHT1;4, they are all expressed exclusively in AM roots (22, 49). In addition, it is reported that the subgroup PHT1;4 proteins LjPT4 and MtPT4 also mediate early root responses to $P_i$ status in nonmycorrhizal roots (21). In contrast, little is known about the functions of subgroup PHT1;8 members. In *A. sinicus*, knockdown of subgroup PHT1;8 gene *AsPT1* by RNAi results in a degenerating arbuscule phenotype, which is similar to that of *AsPT4* RNAi lines, whereas AsPT1 is not required for symbiotic $P_i$ uptake (23). In poplar, PtPT8 and PtPT10, phylogenetically related to subfamily I (Fig. 1), are functionally different; only PtPT10 was specifically induced in AM roots (38). Moreover, *MtPT8*, a second AMF-specific $P_i$ transporter gene of subgroup PHT1;8 in *M. truncatula* (Fig. 1), was

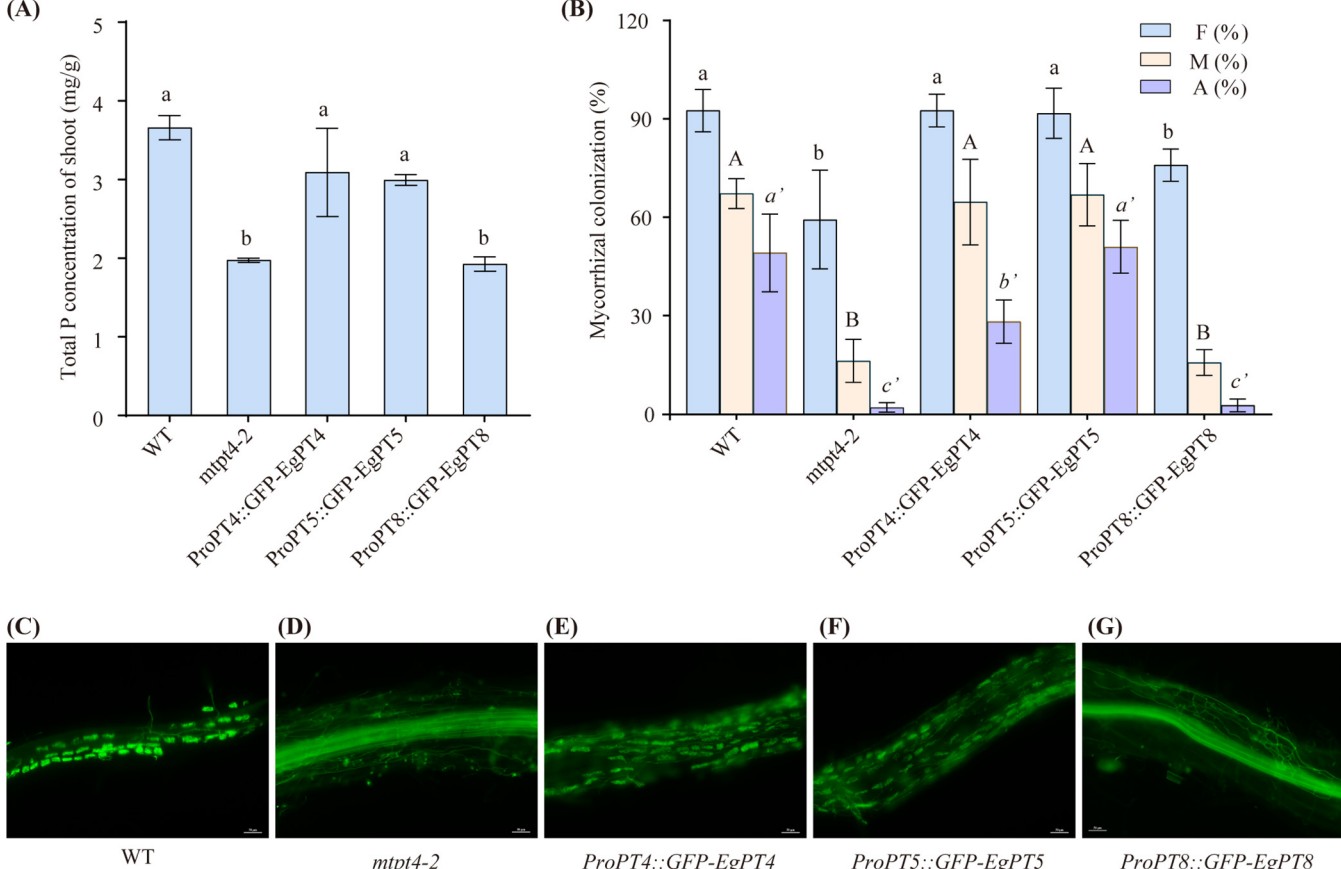

**FIG 7** *EgPT4* and *EgPT5* but not *EgPT8* can complement the defective AM phenotype and mycorrhizal P uptake of the *mtpt4-2* mutant. ProEgPT4::eGFP::EgPT4, ProEgPT5::eGFP::EgPT5, and ProEgPT8::eGFP::EgPT8 were expressed under the control of the native promoter in the *mtpt4-2* mutant plants, respectively. The transgenic lines, as well as the corresponding wild-type (WT) and *MtPT4-2* mutant seedlings, were inoculated with *R. irregularis* and cultured under low $P_i$ (30 $\mu$M) supply conditions for 6 weeks. (A) Total P concentration of shoot. (B) Mycorrhizal colonization of total mycorrhizal frequency (F%), mycorrhizal intensity (M%), and arbuscule abundance (A%) in mycorrhizal roots of these materials were determined. (C to G) Arbuscule morphology was analyzed in WT, *mtpt4-2* mutant, and transgenic lines. Mycorrhizal roots were stained with WGA488. Bars, 20 $\mu$m (a to e) and 50 $\mu$m (f to j). Turkey's test was used for multiple comparison analyses. Error bars indicate SD ($n$ = 3). Different letters denote statistical differences.

highly expressed in low $P_i$ treatment (20 $\mu$M), while *MtPT4* had decreased expression under similar conditions (50). However, in our study, EgPT4 and EgPT5 could restore the phenotype of mutant *mtpt4-2*, and *ProEgPT8::eGFP::EgPT8* lines did not show arbuscular morphology similar to that of wild plants, demonstrating that subfamily I proteins in *Eucalyptus* also have different functions (Fig. 7). This indicates that regulation of subgroup PHT1;8 genes expression in different AM plant species may differ; therefore, we speculate that they will not affect the symbiotic efficiency of host-fungal combinations. Taken together, the functions of the proteins in subgroups PHT1;4 and PHT1;8 are different, but the specific functions of the proteins in subgroup PHT1;8 in response to the AM fungi remain to be elucidated.

**EgPT8 is localized in AM fungi-colonized cortical cells.** So far, in different plant species, all the functionally characterized subfamily I members were confirmed to locate on PAM (12, 23). Although eucalypts evolve in a mutualistic endosymbiosis with AM and ECM fungi, *EgPT8* was specifically expressed in AM root tissues (Fig. 2A to D), similar to the expression model of *PtPT10* in poplar (38), indicating that the expression patterns of subfamily I proteins are conserved in AM symbiosis. Intriguingly, unlike the localization pattern of other AM-specifically induced proteins, such as OsNPF4.5 (41), the subcellular localization analysis in tobacco leaves showed that EgPT8 was located in the ER (Fig. 2E and F). Therefore, we speculate that in cells that cannot form arbuscules, the AM-specific proteins may not be recruited to the PAM. This result was consistent with the localization of MtPT4 in *N. benthamiana*, due to the specific secretory

system resulting in the polar targeting of EgPT8 in the uncolonized root cells (51). Further exploration of the localization of EgPT8 in the yeast cells suggests that it is a PM-localized $P_i$ transporter (Fig. 3C), identical to the localization of AsPT1 and AsPT4, two subfamily I proteins expressed in AM *A. sinicus* (23). In addition, the observation that EgPT8 localized in AM fungi-colonized cortical cells suggests it is a protein that probably acts in the PAM.

**EgPT8 is not required for the $P_i$ uptake pathway in the AM fungi symbiotic systems.** P is a major essential nutrient and is the major limiting factor for plant growth, especially for fast-growing species like eucalypts (52, 53). A growing body of evidence points to the fact that subgroup PHT1;4 members, such as MtPT4 and AsPT4, can take up $P_i$ through the PAM (12, 23); however, the P uptake contribution of subgroup PHT1;8 $P_i$ transporters in AM symbiosis is still unclear. In this study, we revealed that EgPT8 cannot complement the growth defect of the EY917 yeast mutant when supplied only with inorganic phosphate (Fig. 3A). In contrast, under conditions that provide glycerol-3-phosphate and $KH_2PO_4$, EgPT8 can restore the growth phenotype of EY917 (Fig. 3B), identical to the function of OsPT3 in the EY917 cells (54). However, further experimentation determined that the total P concentration in root and shoot tissues in the *RNAi-EgPT8* lines was similar to that in control plants, demonstrating that EgPT8 is not involved in the AM uptake pathway (Fig. 4C and D). Similarly, in *A. sinicus*, AsPT1 encodes a PM-localized transporter that is responsible for the $P_i$ uptake in the yeast mutant strain MB192; however, it is not necessary to regulate the symbiotic $P_i$ uptake (23). Furthermore, by expressing *EgPT4*, *EgPT5*, and *EgPT8* in the *mtpt4-2* mutant, we confirmed that *EgPT4* and *EgPT5* were able to fully complement mycorrhizal $P_i$ uptake of the *mtpt4-2* mutant, whereas EgPT8 was not capable of taking up $P_i$ in the corresponding transgenic plants (Fig. 7A). These data suggest that EgPT8 has no $P_i$ transport activity and thus is not responsible for symbiotic $P_i$ uptake in *E. grandis*.

**EgPT8 is essential for arbuscule elongation, while EgPT4 and EgPT5 are important for arbuscular formation in *E. grandis* inoculated with AM fungi.** Several subfamily I members were functionally verified to be associated with the formation of arbuscules, such as MtPT4 and OsPT11 (21, 22). In our study, the intraradical structures and arbuscular phenotype were normal in *RNAi-EgPT8* lines as *EgPT8* is knocked down (Fig. 6A), which seems contradictory to the findings that knockdown/knockout of the AM-specific upregulated Pht1 genes in *Lotus japonicus*, *M. truncatula*, and *A. sinicus* significantly impaired the development of AM interaction (21, 23, 55). It is likely that the expression of AM-associated genes in *E. grandis* plant and polyP metabolism-related genes in *R. irregularis* was not influenced when the function of *EgPT8* was knocked down (Fig. 5B to E; Fig. S8). Furthermore, well developed arbuscules in the *mtpt4-2* plants expressing *EgPT8* (ProPT8::GFP-EgPT8) were not observed, which differs significantly from the WT plants, (Fig. 7C to G), suggesting that *EgPT8* is not an essential $P_i$ transporter gene that involved in the AM formation in *E. grandis*. In addition, heterogeneous expression of *EgPT4* and *EgPT5* in the *mtpt4-2* mutant could fully complement its defects in AM development and mycorrhizal $P_i$ uptake, further suggesting functional differentiation between the three AM-specific $P_i$ transporters in *E. grandis*. However, unlike the function of homologous gene *AsPT1* (23), the length and area of mature arbuscules in *EgPT8-RNAi* lines were significantly affected compared with the control lines (Fig. 6B and C), demonstrating that *EgPT8* may play a key role in the later stages of cell development, especially in the elongation of mature arbuscules. Thus, combined with the fact that heterogeneous expression of *EgPT4* and *EgPT5* in the *mtpt4-2* mutant could fully complement its defects in AM development (Fig. 7C to G), it could be inferred that the function and regulatory mechanisms of the three AM-specific PHT1 proteins in response to AM symbiosis might be nonredundant in *E. grandis*.

In conclusion, for a better understanding of the symbiotic response of *E. grandis* to AM fungus *R. irregularis* and, especially, deciphering the role of EgPT8 as a major $P_i$ transporter to $P_i$ uptake and AM symbiosis, functional characterizations were conducted through combining the heterologous expression in yeasts, subcellular localization studies, and the reverse genetics approaches during the *in planta* phase. This study demonstrated that the subgroup PHT1;8 transporter EgPT8 has functions in the

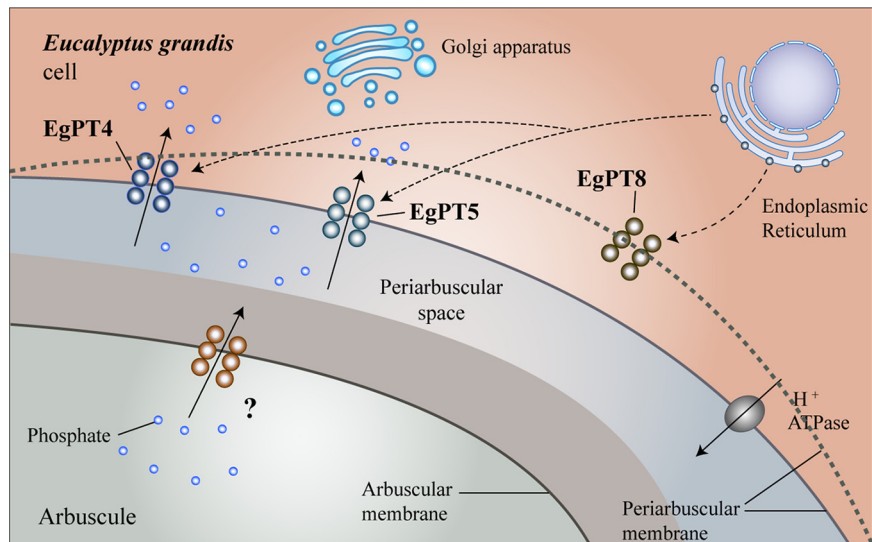

**FIG 8** A model of EgPT8 in AM symbiosis. The EgPT8 $P_i$ transporter engages in elongation of mature arbuscule on the periarbuscular membrane (PAM). In an uninfected cell, EgPT8 is retained in the endoplasmic reticulum. During arbuscule development, EgPT8 is recruited at the symbiotic interface. EgPT4 and EgPT5 are responsible for $P_i$ transport at the symbiotic interface, while EgPT8 is essential to arbuscule elongation. The gray dashed line indicates the extension of PAM.

development of mature arbuscules. Importantly, EgPT8 is not involved in the symbiotic $P_i$ uptake pathway (Fig. 8). Our research provides new insights into the functions of AM-specific $P_i$ transport in the symbiotic system.

## MATERIALS AND METHODS

**Plant materials and mycorrhizal fungi strains, as well as the culture condition.** *E. grandis* seeds, provided by the Research Institute of Tropical Forestry (China Academy of Forestry), were surface sterilized in 2% sodium hypochlorite solution for 10 to 15 min; subsequently shaken and washed five times with double-distilled water (ddH$_2$O); and germinated in a growth chamber programmed for 16 h light at 28°C and 8 h dark at 18°C for 1 week. Then, 1-week-old seedlings were transplanted to plastic pods (20 cm × 15 cm × 3 cm) filled with sterilized quartz sand and irrigated with half-strength MS solution twice a week for a month.

After a month of cultivation, seedlings were then transferred to 9 cm × 9 cm × 7.9 cm pots for inoculation with *R. irregularis*. To guarantee high mycorrhizal colonization, the modified Long-Ashton (mLA) nutrient solution (56) containing 30 $\mu$M NaH$_2$PO$_4$ was fertilized twice a week. For each pot, the seedling was inoculated with approximately 200 spores of *R. irregularis* in the sterilized quartz sand around the roots.

The ECM fungus *S. bovista* was cultured according to Plett et al. with some modifications (57). Briefly, small squares (0.5 cm × 0.5 cm) of *S. bovista* were excised from the leading edge of a 20-day-old colony growing on full-strength modified Melin-Norkrans (MMN) media (1.0% glucose). Then, these inoculums were placed in a culture flask (67 × 67 × 91 mm³) filled with half of the vermiculite, and the MMN solution was added to adjust the humidity to 70%. After 20 days of cultivation, 20 g of inoculum was inoculated to a 6-month-old *E. grandis* seedling. After 3 months of growth, the symbiotic presentation was detected, and ECM *E. grandis* roots were used to extract RNA.

**RNA extraction and expression assay.** The cetyltrimethylammonium bromide (CTAB) method is used for obtaining RNA from 100 mg of tissues (58). For cDNA preparation, 1 mg of total RNA from each sample was used to synthesize first-strand cDNA using the HiScript II first-strand cDNA synthesis kit (Vazyme, Nanjing, China). Quantitative RT-PCR (qRT-PCR) analysis was performed on the CFX Connect real-time PCR detection system (Bio-Rad, USA, 1855200) using the ChamQ Universal SYBR qPCR Master Mix (Vazyme). All qRT-PCR analyses were performed using three biological replicates and three technical replicates. All primers are listed in Table S1.

**Yeast manipulations.** To construct *EgPT4*, *EgPT5*, and *EgPT8* yeast expression vectors, the coding sequences were amplified and cloned into the pFL61 vector via the NotI site. The constructs were transformed into the yeast mutant strain EY917 (MAT$\alpha$ *pho84::HIS3 pho87::CgHIS3 pho89::CgHIS3 pho90::CgHIS3 pho91::KlURA3*) according to Gietz and Schiestl (59). The $P_i$ transporter gene *EgPT4* transformed into EY917 and the empty vector transformed into wild-type strain EY57 (MAT$\alpha$ *ade2-1 trp1-1 can1-100 leu2-3,112 his3-11,15 ura3*) were used as positive controls, while the empty vector transformed into EY917 was used as a negative control. The positive transformants were screened on the synthetic dropout (SD/−Ura) medium plate. The yeast cells were inoculated into SD/−Ura medium (pH 5.8) with 1 mM

glycerol-3-phosphate until $OD_{600}$ = 1. The yeast cells were centrifuged and resuspended in $P_i$-free SD/−Ura medium. After 4 h of starvation treatment, the yeast cells were resuspended again and adjusted to an $OD_{600}$ of 0.5. Subsequently, 5-$\mu$L yeast aliquots of the 10-fold serial gradient were spotted onto SD (−Ura) plates containing different $P_i$ concentrations as indicated in Fig. 3 for growth assays. For the analysis of subcellular localization, the full-length cDNA of *EgPT4*, *EgPT5*, and *EgPT8* was amplified and cloned into expression vector pUG36 via two enzyme sites EcoRI and SalI and was performed in the yeast train EY57. When the value of $OD_{600}$ = 1, 50 mL yeast cells were collected and washed three times with ddH$_2$O and then freeze-dried for total P concentration assay.

**Subcellular localization and heterogeneous expression analysis.** For subcellular localization studies in *N. benthamiana*, the full-length coding sequences of *EgPT4*, *EgPT5*, and *EgPT8* were amplified and recombined into the pCanG-N vector via BamHI restriction site using the ClonExpress II one-step cloning kit (Vazyme Biotech, Nanjing, China). Only the pCanG-N-EgPT4 vector was transformed into *A. tumefaciens* strain GV3101 for subcellular localization analysis; pCanG-N-EgPT5 and pCanG-N-EgPT8 were used to construct heterogeneous expression vectors. *Agrobacterium*-mediated transformation of *N. benthamiana* was performed as described previously (60).

For subcellular localization assay in *M. truncatula* and heterogeneous expression analysis, the full-length cDNA sequences of *EgPT4*, *EgPT5*, and *EgPT8* with the stop codon and fused with the GFP reporter gene were amplified from the pCANG-N-EgPT8 vector and cloned into the modified binary vectors pCAMBIA1305 by using the ClonExpress II one-step cloning kit (Vazyme Biotech, Nanjing, China). Then, to replace the 35S promoter in front of the GFP-EgPT4, GFP-EgPT5, and GFP-EgPT8 chimeric genes, 1,293-, 1,378-, and 1,381-bp-long promoter fragments of *EgPT4*, *EgPT5*, and *EgPT8* were amplified and cloned into the KpnI and EcoRI restriction sites of reconstructed pCAMBIA1305 vector, respectively. The transformation of *M. truncatula* A17 was performed as described by Limpens et al. (61), while the *A. rhizogenes* strain we used was MSU440.

**Construction of binary vector for EgPT8 interference and transformation of *E. grandis*.** For the construct of *EgPT8-RNAi*, a 229-bp fragment of *EgPT8* was amplified by specific primers and subcloned into an intermediate vector pDONR221 according to the Gateway recombination technology (Invitrogen) and then recombined into the destination vector pK7GWIWG2D(II)-RootRed (43). The above construct and empty vector (Cheap-RNAi) were then transformed into the *A. rhizogenes* strain A4RS.

The plant transformation of *E. grandis* was performed according to Plasencia et al. with some modifications (62). Briefly, the seeds of *E. grandis* were germinated on one-quarter-strength MS medium solidified with 0.3% Phytagel (Sigma). Then, the base of hypocotyls of 14-day-old seedlings was cut and dipped in A4RS liquid culture for 10 min. After being dried on the sterilized filter paper, the seedlings were placed on half-strength MS medium supplemented with 30 g/liter of sucrose and 1 mL/liter acetosyringone at 25°C in the dark for 1 day. Then, cocultures were transferred to a chamber programmed for 16 h light (7 $\mu$mol/m$^2$s) and 8 h dark, 50% humidity, at 20 to 25°C for another 13 days. To prevent the growth of A4RS, the plants were then transferred to a new half-strength MS medium without acetosyringone and containing 250 mg/liter cephalosporin, and the light condition was changed to 12 $\mu$mol/m$^2$s. In this stage, the DsRed signal of transformed plants was screened using a fluorescence stereomicroscope (Nikon SMZ18).

**Total P concentration assay.** For the measurement of total P concentration in different samples of the plant tissues and yeast cells, ~0.05 g dry ground powder was used following the method described by Fan et al. (63).

**Microscopy and cell imaging.** Fluorescence microscopy images of mycorrhizal *M. truncatula* roots were photographed using fluorescence microscopy (Nikon, Y-TV55). The fluorescence stereomicroscope (Nikon, DS-R12) was used to detect the hairy roots with red fluorescence. Confocal microscopy images of subcellular localization and AM roots phenotype analysis were taken by the laser-scanning confocal microscopy (Zeiss 780), equipped with long-distance ×63 water and ×100 oil immersion objectives. Excitation/emission wavelengths were 488/495 nm to 556 nm for GFP and 543/565 nm to 615 nm for mCherry.

**Statistical analysis.** All the data collected were analyzed using IBM SPSS Statistics 22 program (SPSS, Chicago, IL, USA). Statistical analyses were performed by Student's *t* test. Turkey's honestly significant difference (HSD) test was used for multiple comparison analysis. All data are shown as the means ± SEM of three independent replicates. The significance of differences is marked as follows: *, $P < 0.05$; **, $P < 0.01$; and ***, $P < 0.01$, or by different letters ($P < 0.05$). Analysis of arbuscule width and area is based on software ImageJ (https://imagej.nih.gov/ij/).

**Bioinformatic analysis.** The BLASTp was used to search for homologs EgPT8 protein in the plant species (http://blast.ncbi.nlm.nih.gov/Blast.cgi). Structure alignment between the crystal structure of MtPT4 and the model structure of EgPT4, EgPT5, and EgPT8 was analyzed using SWISS-MODEL (https://swissmodel.expasy.org/) and PyMOL (version 2.5.2).

**Phylogenetic analysis.** The phylogenetic analysis was performed using protein sequences of PHT1 homologs by the neighbor-joining algorithm within the MEGA 7.0 program. The evolutionary distances were computed using the Poisson correction method. Bootstrap analysis was carried out with 1,000 replicates. The reference numbers of the protein sequences used for constructing the tree are provided in Table S2 in the supplemental material.

**Data availability.** Sequence information from this paper can be searched in GenBank libraries under the following accession numbers for PHT1 family proteins: *E. grandis* EgPT4 (ON012824), EgPT5 (ON012825), and EgPT8 (ON012828); *Populus trichocarpa* PtPT8 (XP_024446611.1) and PtPT10 (XP_006374329.2); *L. japonicus* LjPT4 (BAG71408.1) and LjPT8 (Lj2g3v2172990.1); *M. truncatula* MtPT4 (XP_013466381.1) and MtPT8 (XP_003615445.1); *Glycine max* GmPT11 (NP_001239765.1) and GmPT13 (NP_001241400.1); *Solanum lycopersicum* SlPT4 (NP_001234674.2) and SlPT5 (XP_004240951.1); *Solanum tuberosum* StPT4

(XP_006360181.1) and StPT5 (XP_006360180.1); *A. sinicus* AsPT1 (AFU50500.1) and AsPT4 (AFU50504.1); and *O. sativa* OsPT11, (AAN39052.1). Detailed information is provided in Table S2 in the supplemental materials.

## SUPPLEMENTAL MATERIAL

Supplemental material is available online only.

**SUPPLEMENTAL FILE 1**, PDF file, 1.8 MB.

## ACKNOWLEDGMENTS

This work was supported by grant 201904020022 from the Key Projects of Guangzhou of Science and Technology Plan, grant NZ2021025 from the Laboratory of Lingnan Modern Agriculture Project, grant 2018A030313141 from the Natural Science Foundation of Guangdong Province in China, and grants 31800092 and 32071639 from the National Natural Science Foundation of China.

We declare no conflicts of interest.

The authors are grateful to An Jianyong (Huazhong Agricultural University, Wuhan, China) for kindly providing gateway vectors and mutant *mtpt4-2* for this study, to Chunmei Zhong (South China Agricultural University, Guangzhou, China) for kindly providing the plasmid pCanG-N for transient expression analysis, and to Aiqun Chen (Nanjing Agricultural University, Nanjing, China) for kindly providing the binary vector pCAMBIA1305. We thank Luisa Lanfranco and Veronica Volpe (University of Turin, Turin, Italy) for kindly providing the pFL61 vector and EY57/917 strains for the yeast expression, respectively.

M.T., H.C., and X.X. designed the experiments and managed the projects. X.C. and W.S. performed experiments. X.C. and X.W. performed data analysis. S.W., Y.R., L.H., and W.L. performed experiment of plant management. X.C. and M.T. wrote the manuscript. C.H. and H.W. assisted with the interpretation of the results. All authors have read, edited, and approved the current version of the manuscript.

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
