## [Reviewer comments · Microbiology Spectrum]

Microbiology Spectrum

A Eucalyptus Pht1 Family Gene EgPT8 is Essential for Fungal Arbuscule Elongation of Rhizophagus irregularis

Xianrong Che, Sijia Wang, Ying Ren, Xianan Xie, Wentao Hu, Hui Chen, and Ming Tang

Corresponding Author(s): Xianrong Che, South China Agricultural University

Review Timeline:

Submission Date:	April 26, 2022
Editorial Decision:	July 18, 2022
Revision Received:	September 17, 2022
Accepted:	September 22, 2022

Editor: Jeffrey Gralnick

Reviewer(s): The reviewers have opted to remain anonymous.

Transaction Report:

DOI: <https://doi.org/10.1128/spectrum.01470-22>

July 18, 2022

Dr. Xianrong Che
South China Agricultural University
College of Forestry and Landscape Architecture
Guangzhou
China

Re: Spectrum01470-22 (A *Eucalyptus* Pht1 Family Gene *EgPT8* is Essential for Fungal Arbuscule Elongation of *Rhizophagus irregularis*)

Dear Dr. Xianrong Che:

Thank you for submitting your manuscript to Microbiology Spectrum. Two experts in the field have reviewed your manuscript and have provided important feedback that you can use to revise your manuscript. In addition to addressing each of the reviewer comments, be sure to pay special attention to the appropriateness of statistical tests and how statistics are reported. You should also carefully check grammar and word usage throughout the manuscript as both authors noted some issues there.

Link Not Available

Sincerely,

Benjamin Wolfe

Journals Department
Reviewer comments:

Reviewer #1 (Comments for the Author):

The manuscript "A *Eucalyptus* Pht1 Family Gene *EgPT8* is Essential for Fungal Arbuscule Elongation of *Rhizophagus irregularis*" by Xianrong Che and colleagues describes the functional characterization of the phosphate transporter *EgPT8* in mycorrhizal roots of eucalyptus. *EgPT8* is a member of a phylogenetic clade that does not contain the well-characterized AM PTs, therefore its investigation is very interesting. Surprisingly, although *EgPT8* can function in yeast Pi uptake (to some degree)

and its localization strongly resembles that of previously characterized AM PTs, EgPT8 does not appear to have a function in symbiotic Pi uptake via arbuscules.

While the paper does not solve the question of how PHT1:4 and PHT1:8 differ, it shows that they have different roles during AM symbiosis. This paper is interesting and opens multiple interesting lines of research. However, some data representations/interpretations are a bit confusing. Please see below for details.

Title: consider deleting the word 'fungal'

Line 139/140: the sentence "to examine the sensitivity of EgPT8 expression to AM roots of *E. grandis*" is a bit confusing. Shouldn't it say "to determine the expression level of EgPT8 in *E. grandis* roots at different time points of AM symbiosis"?

Line 153 ff: The authors state in the text that GFP-EgPT8 co-localizes with ER marker in *N. benthamiana* leaves. Looking at Fig S4, I am not so sure this conclusion is necessarily biologically relevant. To me it seems like there may also be at least partial co-localization with the plasma membrane marker. In our experience, *N. benthamiana* over-expression often results in ER accumulation of fusion proteins, but this may simply be due to very high gene expression. I am not sure we can conclude that "EgPT8 were resident in the ER of *N. benthamiana* leaf epidermal cells in nonmycorrhizal tissue" (line 155/156).

The fluorescence image in Fig 2 is more informative since it uses the endogenous EgPT8 promoter and the tissue in which the protein normally is expressed (roots). However, although these images look very much like the PAM-localization that has been previously shown for e.g. Medicago PT4 and other PTs, I feel the sentence "This evidence indicates that EgPT8 is exclusively localized to the PAM" (line 163/164 and also 170) is not warranted as co-localization of EgPT8 to the PAM was not shown (and sub-cellular localization was concluded to be in the ER). This experiment has to be performed or the sentence and all PAM-related sentences throughout the whole manuscript need to be tuned down.

Line 189 ff: The sentence "the growth rate of mutant EY917 carrying the pFL61-EgPT8 is significantly higher than that of EY917 cells with pFL61 at the logarithmic phase (from 8 to 20 h; Fig. 3D), however, is lower than EgPT4 and EgPT5 expressing in EY917" does not make sense with what is shown in Figure 3D. If I interpret the figure correctly, PT4 is higher but PT5 is lower than PT8?

Also, there are no statistical tests shown in Fig. 3 or the text, so the word "significantly" is not justified.

Line 210: consider re-wording ('inactivated effect of ...').

Line 224: AsPT1 is not involved in symbiotic Pi uptake

Figure 1: it would be helpful for the reader to highlight the eucalyptus PTs discussed in this manuscript.

Figure 2B: it would be helpful to see some statistics on difference in gene expression level at the different time points.

Figure 2D: I am a bit confused. The text (line 148) and the figure legend indicate this is the expression level of EgPT8 when inoculated with *S. Bovista*, but the figure seems to show EgPT4 gene expression? Please clarify.

Figure 3C: Bottom panel lacks annotation.

Figure 3E: What do the significance asterisks refer to? According to the figure legend, a Tukey test was performed but it is unclear which of the pairwise comparisons is shown. This is by the way the case for many of the main and supplementary figures.

Figure 4: Figure legend indicates samples were analyzed with student's t-test. This test is only applicable when comparing 2 samples. When more than 2 samples are being analyzed, an anove test should be performed. Which sample was the control for the t-test?

Figure 6A g-l: It would be very helpful to scale these images to the same size. The way it is represented now, it is impossible for the reader to inspect the arbuscule sizes.

Figure S4: Panel with mCherry/DsRed images says 'brightfield'.

Reviewer #2 (Comments for the Author):

The submitted paper by Che and colleagues considers the role of the PT1:8 family of putative phosphate transporters, with a special focus on EgPT8. They undertake a range of in silico and experimental approaches to advance our understanding of this gene during AM symbioses. Their research models are varied and appropriate for the work being undertaken. The paper is generally well written with solid results and a nice interpretation of the work without over-extending the conclusions. This paper

contributes nicely to the field. I only have a few minor editorial comments:

Line 53: "Symbionts" should be "symbioses"

Line 105-106: you need to reference this "previous" work

Line 221: This is a complicated sentence due to the double negative. It would be clearer just to say that "EgPT8 is dispensable...."

Line 225, 230: remove "as expected"

Line 286: "eucalypts" should be "eucalypt"

Line 297: "arbuscule" should be plural

Line 331: "Token" should be "taken"

Staff Comments:

Preparing Revision Guidelines

Please return the manuscript within 60 days; if you cannot complete the modification within this time period, please contact me. If you do not wish to modify the manuscript and prefer to submit it to another journal, please notify me of your decision immediately so that the manuscript may be formally withdrawn from consideration by Microbiology Spectrum.

Dear Editor and Reviewers,

We would like to thank you for your careful reading, helpful comments, and constructive suggestions, which has significantly improved the presentation of our manuscript. Based on the instructions provided in your letter, we uploaded the file of the revised manuscript. Accordingly, we have uploaded a copy of the original manuscript with all the changes highlighted by using the track changes mode in MS Word. We have carefully considered all comments from the reviewers and revised our manuscript accordingly. In the following section, we summarize our responses to each comment from the reviewers. The comments are reproduced and our responses are given directly afterward. We explain the appropriateness of statistical tests and how statistics are reported. We would like also to thank you for allowing us to resubmit a revised copy of the manuscript. We believe that our responses have well addressed all concerns from the reviewers. We hope our revised manuscript can be accepted for publication. We hope that the revised manuscript is accepted for publication in the Microbiology Spectrum.

Sincerely

Ming Tang

Reviewer #1

Comment 1: *Title: consider deleting the word 'fungal'*

Reply 1: Thanks for the reviewer's kind suggestion. We have deleted the word 'fungal' from the title (Please see Line 1).

Comment 2: *Line 139/140: the sentence "to examine the sensitivity of EgPT8 expression to AM roots of E. grandis" is a bit confusing. Shouldn't it say "to determine the expression level of EgPT8 in E. grandis roots at different time points of AM symbiosis"?*

Response 2: Thank you for your significant reminding. We have changed this word

according your suggestion. The revised details can be found in Line 139-141.

Comment 3: *Line 153 ff: The authors state in the text that GFP-EgPT8 co-localizes with ER maker in N. benthamiana leaves. Looking at Fig S4, I am not so sure this conclusion is necessarily biologically relevant. To me it seems like there may also be at least partial co-localization with the plasma membrane marker. In our experience, N. benthamiana over-expression often results in ER accumulation of fusion proteins, but this may simply be due to very high gene expression. I am not sure we can conclude that "EgPT8 were resident in the ER of N. benthamiana leaf epidermal cells in nonmycorrhizal tissue" (line 155/156).*

The fluorescence image in Fig 2 is more informative since it uses the endogenous EgPT8 promoter and the tissue in which the protein normally is expressed (roots). However, although these images look very much like the PAM-localization that has been previously shown for e.g. Medicago PT4 and other PTs, I feel the sentence "This evidence indicates that EgPT8 is exclusively localized to the PAM" (line 163/164 and also 170) is not warranted as co-localization of EgPT8 to the PAM was not shown (and sub-cellular localization was concluded to be in the ER). This experiment has to be performed or the sentence and all PAM-related sentences throughout the whole manuscript need to be tuned down.

Reply 3: Thank you for your comment. Our result of membrane localization of EgPT8 is consistent with previous report that the AM-specific proteins is able to polarly localize to the PAM upon AM symbiosis to facilitate the symbiotic Pi uptake (Pumplina et al., 2012). However, in the tobacco leaves or other plant tissues where cannot form the AM symbiotic system, resulting in the polar targeting of EgPT8. Once arbuscules are formed in the tissue, the AM-specific proteins will be recruited to the correct location (Che et al., 2022). Further, the full-length sequences of 17 of the 22 Pi transporters of *E. grandis* were isolated from different tissues, in which the relative expression levels of two genes were higher in root tissues than other *E. grandis* PHT1 genes including EgPT4, EgPT5 and EgPT8. However, the two genes

were all co-localized with the plasma membrane marker (The data have not been published).

Moreover, we have also presented subcellular localization of the EgPT8 transporter in yeast cells, our result demonstrated that EgPT8 protein is targeted to the plasma membrane of yeast cells (Fig. 3C). According to your advice, all PAM-related sentences throughout the whole manuscript have been revised. We hope that our data is sufficient to give the conclusion that EgPT8 is localized in AM fungi-colonized cortical cells.

Pumplina, N., Zhang, X., Noara, R.D., and Harrison, M.J. (2012) Polar localization of a symbiosis-specific phosphate transporter is mediated by a transient reorientation of secretion. *Proc. Natl. Acad. Sci. USA* 109 (11): E665-E672.

Che, X., Lai, W., Wang, S., et al. (2022) Multiple PHT1 family phosphate transporters are recruited for mycorrhizal symbiosis in *Eucalyptus grandis* and conserved PHT1;4 is a requirement for the arbuscular mycorrhizal symbiosis. *Tree Physiology*. <https://doi.org/10.1093/treephys/tpac050>.

Comment 4: *Line 189 ff: The sentence "the growth rate of mutant EY917 carrying the pFL61-EgPT8 is significantly higher than that of EY917 cells with pFL61 at the logarithmic phase (from 8 to 20 h; Fig. 3D), however, is lower than EgPT4 and EgPT5 expressing in EY917" does not make sense with what is shown in Figure 3D. If I interpret the figure correctly, PT4 is higher but PT5 is lower than PT8?*

Also, there are no statistical tests shown in Fig. 3 or the text, so the word "significantly" is not justified.

Reply 4: We appreciate for your valuable comment. We have revised the text to address your concerns and hope that it is now clearer. Please see Lines 188-192 of the revised manuscript. The revised details are as follows:

Next, we introduced the pFL61-EgPT8 into EY917 to test the growth tendency. In agreement with the complementation of EgPT8 in EY917, the growth rate of mutant EY917 carrying the pFL61-EgPT8 is higher than that of EY917 cells with pFL61 or pFL61-EgPT5 at the logarithmic phase (from 8 to 20 h; Fig. 3D), however, is lower

than EgPT4 expressing in EY917.

Comment 5: *Line 210: consider re-wording ('inactivated effect of ...')*

Reply 5: Thanks for your kind suggestion. The revised details are as follows (lines 209-212):

When the *EgPT8-RNAi* lines were harvested at 49 dpi under low Pi conditions, we identified three lines by qRT-PCR determination, *EgPT8-RNAi-1*, *EgPT8-RNAi-2*, and *EgPT8-RNAi-3*, showing significantly decreased expression levels of *EgPT8* (Fig. 4B).

Comment 6: *Line 224: AsPT1 is not involved in symbiotic Pi uptake*

Reply 6: Thank you for your significant reminding. An earlier study has determined that AsPT1 is required for AM symbiosis, however, AsPT1 is not involved in symbiotic Pi uptake. To demonstrate whether EgPT8 is indispensable for arbuscule formation, we qualified the mycorrhization in the *RNAi-EgPT8* lines and control lines (Please see lines 223-225).

Comment 7: *Figure 1: it would be helpful for the reader to highlight the eucalyptus PTs discussed in this manuscript.*

Reply 7: We appreciate to you for your valuable advice. The eucalyptus PTs in Figure 1 have been marked in red (Please see Fig. 1).

Comment 8: *Figure 2B: it would be helpful to see some statistics on difference in gene expression level at the different time points.*

Reply 8: Thank you for your valuable comment. We reanalyzed the data and marked the significant differences in Figure 2B.

Comment 9: *Figure 2D: I am a bit confused. The text (line 148) and the figure legend indicate this is the expression level of EgPT8 when inoculated with S. Bovista, but the*

figure seems to show EgPT4 gene expression? Please clarify.

Reply 9: We appreciate your valuable comment. We are sorry that this error in the text has bothered you. The revised details can be found in Figure 2D.

Comment 10: *Figure 3C: Bottom panel lacks annotation.*

Reply 10: Thank you for your advice. We have corrected this error. Please see Fig. 3C.

Comment 11: *Figure 3E: What do the significance asterisks refer to? According to the figure legend, a Tukey test was performed but it is unclear which of the pairwise comparisons is shown. This is by the way the case for many of the main and supplementary figures.*

Reply 11: Thank you for pointing out this problem in our manuscript. We are sorry this error in the text has bothered you, it has now been corrected. Please see the figure legend of our revised manuscript. And we point out the control groups of the main and supplementary figures. Please see our revised manuscript.

Comment 12: *Figure 4: Figure legend indicates samples were analyzed with student's t-test. This test is only applicable when comparing 2 samples. When more than 2 samples are being analyzed, an anove test should be performed. Which sample was the control for the t-test?*

Reply 12: Thank you for pointing out this problem in our manuscript. Control-1, Control-2 and Control-3 were set as the reference groups. Significance: * $P < 0.05$, *** $P < 0.001$; Student's *t*-test. Please see lines 809-811.

Comment 13: *Figure 6A g-l: It would be very helpful to scale these images to the same size. The way it is represented now, it is impossible for the reader to inspect the arbuscule sizes.*

Reply 13: Thank you for your valuable comment. Figure 6A shows arbuscular phenotype in control and RNAi lines , while arbuscule sizes were analyzed in Figure

6B-C.

Comment 14: Figure S4: Panel with mCherry/DsRed images says 'brightfield'.

Reply 14: Thank you for your kind advice. We have corrected 'brightfield' to 'mCherry' according your comment. Please see Fig. S4.

Reviewer #2

Comment 1: *Line 53: "Symbionts" should be "symbioses"*

Reply 1: Thank you for your valuable comment. We have corrected this word in our revised manuscript (Please see line 53).

Comment 2: *Line 105-106: you need to reference this "previous" work*

Reply 2: Thank you for your significant reminding. We have provided our previous work (Please see Line 106). The details are as follows:

36. Che, X., Lai, W., Wang, S., et al. (2022) Multiple PHT1 family phosphate transporters are recruited for mycorrhizal symbiosis in *Eucalyptus grandis* and conserved PHT1;4 is a requirement for the arbuscular mycorrhizal symbiosis. *Tree Physiology*. <https://doi.org/10.1093/treephys/tpac050>.

Comment 3: *Line 221: This is a complicated sentence due to the double negative. It would be clearer just to say that "EgPT8 is dispensable...."*

Reply 3: Thank you for your kind advice. The revised details are as follows:

These findings indicate that EgPT8 is dispensable for absorbing Pi from *R. irregularis*.

The revised details can be found in Line 221.

Comment 4: *Line 225, 230: remove "as expected"*

Reply 4: Thank you for your advice. The words 'as expected' have been deleted.

Comment 5: *Line 286: "eucalypts" should be "eucalypt"*

Reply 5: Thanks for your kind suggestion. We reconfirmed the spelling of this word.

Comment 6: *Line 297: "arbuscule" should be plural*

Reply 6: We appreciate for your comment. We have corrected this error.

Comment 7: *Line 331: "Token" should be "taken"*

Reply 7: Thank you for your advice. We have corrected this word in our revised manuscript (Please see Line 330).

September 22, 2022

Dr. Xianrong Che
South China Agricultural University
College of Forestry and Landscape Architecture
Guangzhou
China

Re: Spectrum01470-22R1 (A *Eucalyptus* Pht1 Family Gene *EgPT8* is Essential for Fungal Arbuscule Elongation of *Rhizophagus irregularis*)

Dear Dr. Xianrong Che:

Your manuscript has been accepted, and I am forwarding it to the ASM Journals Department for publication. You will be notified when your proofs are ready to be viewed.

Sincerely,

Jeffrey Gralnick
Editor, Microbiology Spectrum
